# Numerical Simulation of the Frequency Dependence of Fatigue Failure for a Viscoelastic Medium Considering Internal Heat Generation

**DOI:** 10.3390/ma17246202

**Published:** 2024-12-19

**Authors:** Natsuko Kudo, Takumi Sekino, M. J. Mohammad Fikry, Jun Koyanagi

**Affiliations:** 1Department of Materials Science and Technology, Graduate School of Tokyo University of Science, 6-3-1, Niijuku Katsushika-ku, Tokyo 125-8585, Japan; 8219b01@alumni.tus.ac.jp (N.K.); 8220054@alumni.tus.ac.jp (T.S.); 2Department of Mechanical Engineering, The University of Akron, 244 Sumner St., Akron, OH 44325-3903, USA; mfikry@uakron.edu; 3Department of Materials Science and Technology, Tokyo University of Science, 6-3-1, Niijuku Katsushika-ku, Tokyo 125-8585, Japan

**Keywords:** numerical simulation, fatigue failure, viscoelastic, frequency dependence, internal heat generation

## Abstract

Accurately predicting fatigue failure in CFRP laminates requires an understanding of the cyclic behavior of their resin matrix, which plays a crucial role in the materials’ overall performance. This study focuses on the temperature elevation during the cyclic loadings of the resin, driven by inelastic deformations that increase the dissipated energy. At low loading frequencies, the dissipated energy is effectively released as heat, preventing significant temperature rise and maintaining a consistent, balanced thermal state. However, at higher frequencies, the rate of energy dissipation exceeds the system’s ability to release heat, causing temperature accumulation and accelerating damage progression. To address this issue, the study incorporates non-recoverable strain into a fatigue simulation framework, enabling the accurate modeling of the temperature-dependent fatigue behavior. At 0.1 Hz, damage accumulates rapidly due to significant inelastic deformation per cycle. As the frequency increases to around 2 Hz, the number of cycles until failure rises, indicating reduced damage per cycle. Beyond 2 Hz, higher frequencies result in accelerated temperature rises and damage progression. These findings emphasize the strong link between the loading frequency, non-recoverable strain, and temperature elevation, providing a robust tool for analyzing resin behavior. This approach represents an advancement in simulating the fatigue behavior of resin across a range of frequencies, offering insights for more reliable fatigue life predictions.

## 1. Introduction

Carbon-fiber-reinforced plastics (CFRPs) are frequently employed in load-bearing applications, such as aircraft wings and helicopter blades, where a high specific strength and stiffness are critical [1,2,3,4,5]. These structures are often subjected to vibrations and fluctuating loads, leading to fatigue damage [6,7,8,9,10,11]. Notably, fatigue is a major factor contributing to the failure of structural components. While it usually involves the gradual growth of a dominant crack in uniform materials such as metals [12], the fatigue mechanisms in CFRPs are more complex and involve various interacting processes throughout the material [13]. The fatigue behavior of CFRPs, as shown in Figure 1, depends significantly on the frequency of cyclic loadings. At low frequencies, fatigue damage progresses more significantly within each cycle due to the viscoelastic nature of the resin in CFRPs, resulting in fewer cycles before failure [14]. In contrast, at high frequencies, rapid cyclic loading generates heat within the specimen, accelerating damage progression due to elevated temperatures [15]; this again reduces the number of cycles to failure [11,16,17,18,19]. Thus, both extremes—low- and high-frequency cyclic loading—result in a shorter fatigue life in CFRP laminates.

The complexity of fatigue mechanisms in composite materials, including CFRPs, demands a reliable criterion for accurately estimating their lifetime. This has led to the development of various fatigue damage accumulation theories. Among these, the entropy-based approach has gained significant attention due to its ability to quantify irreversible degradation [14,20,21,22,23,24,25]. Entropy generation is calculated by dividing the dissipated energy by temperature and integrating this over the time to fracture [26,27]. This dissipated energy, a product of the stress and inelastic deformation increments, is directly linked to entropy generation [28]. From a thermodynamic standpoint, as systems degrade, entropy rises with the inelastic deformation, ultimately leading to failure. This makes entropy a valuable metric for assessing degradation and predicting failure. Historically, mechanical hysteresis energy has been used to link metal fatigue to cyclic stress–strain behavior [29,30]. Park and Nelson refined this model by incorporating stress states and the elastic strain energy density, applying it to multiaxial fatigue life predictions [31]. Building on these concepts, Bryant et al. further integrated the first and second laws of thermodynamics with the Helmholtz free energy. The authors leveraged the relationship between degradation and entropy generation to predict fatigue life based on parameters such as stress, strain, number of cycles, and time to failure, while also accounting for environmental factors such as temperature and heat [32]. Naderi et al. introduced the concept of fracture fatigue entropy (FFE) as a material-specific criterion for fatigue assessment and showed that it is independent of the load, frequency, or geometry [24,33]. Liakat extended this by validating the FFE model in high-cycle fatigue scenarios, where the anelastic energy at stress levels below the yield strength was measured, confirming the effectiveness of the concept [34].

Recently, researchers have started applying this approach to polymer and composite materials. For instance, Koyanagi et al. [35] used entropy generation as a criterion to simulate transverse cracking in CFRP laminates, modifying Hashin’s failure criterion to account for strength degradation due to entropy. Their simulations, validated by experimental results, showed that damage evolution occurs more rapidly at low frequencies. Similarly, Deng et al. [26] employed entropy generation to model the fatigue behavior of CFRPs via a microscale approach. This model successfully simulated fatigue under multi-amplitude loadings at various stress levels, though it was limited to low-frequency conditions, where heat generation was negligible. In addition, Li et al. [36] explored the impact of the load history on the premature failure of the viscoelastic polymer matrix in CFRPs using an FFE-based method. The study applied the FFE damage criterion under variable-amplitude load modes, including frequent amplitude changes and intermittent pauses, offering a more practical approach for predicting fatigue life under complex loading conditions. However, these models lack the ability to simulate high-frequency loadings, where temperature elevations become a significant factor in material degradation.

In addition to the role of entropy, research has shown that the longevity of composite materials, particularly CFRPs, is heavily influenced by the properties of their resin matrix [37]. The premature failure of the resin matrix, especially under random loadings, is a key factor in determining the lifespan of CFRPs. Combining insights from entropy-based models with a deeper understanding of the resin matrix behavior could offer a more comprehensive framework for predicting fatigue life in these advanced materials. In this study, we propose a novel model that simulates the behavior of epoxy resin, one of the most commonly used resins in CFRPs, under cyclic loadings at both low and high frequencies. The model introduced in our previous study [28], though effective in simulating the behavior at low frequencies, did not account for the effects of higher frequencies. The present model overcomes this limitation by incorporating key parameters such as heat generation, heat release, and non-recoverable strain into the fatigue simulation framework using Abaqus software (Abaqus 2020, Dassault Système). The stress is dynamically updated via a user subroutine that factors in damage, which is calculated based on the temperature and entropy generation—linked to the dissipated energy in the system. The simulation results from the present model align with the experimental data shown in Figure 1, where at low frequencies, the number of cyclic loadings before failure increases as the frequency increases, while at high frequencies, the number of cycles decreases with increasing frequency due to the onset of heat-induced damage. This model represents a significant advancement in fatigue simulation as it can accurately simulate fatigue failure over a wide range of loading frequencies. It therefore offers more precise predictions of the fatigue life of epoxy-based CFRP composites for different loading conditions.

## 2. Numerical Methods

### 2.1. Constitutive Equation Incorporating the Entropy Damage Criterion

As shown in Figure 2, the current model comprises 15 parallel Maxwell elements, each consisting of a spring and a dashpot, along with an additional plastic element.

These 15 Maxwell elements were chosen based on previous studies [27], which demonstrated that this configuration is necessary to accurately reproduce the viscoelastic behavior of the resin. The dashpots, with varying viscosity coefficients, function across multiple timescales. For shorter timescales, dashpots with smaller viscosity coefficients primarily govern the response, while for longer timescales, dashpots with larger viscosity coefficients dominate. This configuration allows the 15 independent elements to collectively represent a broad spectrum of material behaviors. The plastic element, absent from the previous model [26,27], plays a crucial role in calculating the temperature elevation by accounting for the material’s frequency-dependent behavior.

Resin deformation comprises both elastic and inelastic components, with only the inelastic deformations contributing to heat generation. Prior studies [26,27] have modeled the inelastic deformations using moving dashpots but have only considered differences in the stress levels. To address the frequency dependence, this study introduces a plastic element, as given by Equation (1):(1)$$\Delta \varepsilon^{nr}\left(i\right)=\left(a\times {\sigma (i)}^{c}\exp\left(-{\left(\frac{\sigma (i)}{\sigma_{0}}\right)}^{d}\right)+b\left(1-\exp\left(-{\left(\frac{\sigma (i)}{\sigma_{0}}\right)}^{d}\right)\right)\right)\times \Delta \varepsilon \left(i\right),$$
where $\Delta \varepsilon^{nr}$ and Δε represent the non-recoverable and total strain increments, respectively; and the parameters a, b, c, and d are fitting constants. The non-recoverable strain increment is determined by the stress in each direction, where the first term is proportional to the cth power of the stress, and the second term is multiplied by a constant, b. These two terms are connected through the constant $\sigma_{0}$, which is combined based on the stress. Using Equation (1), the relationship between stress and the non-recoverable strain increment, normalized by the total strain increment, is illustrated in Figure 3. At low stress levels, the non-recoverable strain increment is minimal; as the stress increases, it grows and eventually reaches a constant ratio of the total strain increment. This approach aims to simulate frequency-dependent behavior.

The stress is determined by summing all the elastic components, with stress relaxation occurring independently for each element. The viscoelastic constitutive law incorporating the damage effects can be formulated as in Equation (2) [38]:(2)$$\sigma_{ij}\left(t\right)=\left(1-D\right)\int_{0}^{t}E_{ijkl}^{r}\left(t-t^{\prime }\right)g\frac{d\varepsilon_{kl}^{ve}}{dt^{\prime }}dt^{\prime },$$
where the relaxation modulus, $E_{ijkl}^{r}$, is defined as in Equation (3):(3)$$E_{ijkl}^{r}\left(t\right)=\sum_{n=1}^{15}E_{ijkl}^{n}e^{-\frac{tE^{n}}{\eta^{n}}},$$

Here, D, $\varepsilon_{kl}^{ve}$, g, and t represent the damage variable, viscoelastic strain, nonlinear coefficient, and time, respectively. The relaxation modulus, $E_{ijkl}^{r}$, was computed using the spring, $E^{n}$, and the dashpot, $\eta^{n}$ (n=1, 2, ..., 15). The nonlinear coefficient, g, calculated using Equation (4), is related to the von Mises stress, $\sigma_{Mises}$, and the specific stress, $\sigma_{0}$.
(4)$$g=\frac{1}{1+\alpha {\left(\frac{\sigma_{Mises}}{\sigma_{0}}\right)}^{m}},$$

The model employs entropy generation as the criterion for damage assessment. Previous studies [27] have concluded that material failure occurs when the accumulated entropy reaches the FFE threshold. The FFE, $s^{cr}$, remains constant, irrespective of the loading level, number of cycles, or loading frequency. Since the entropy generation, s, primarily results from inelastic deformations, which is calculated using Equation (5).
(5)$$s=\int_{0}^{t}\frac{\sigma :\dot{\varepsilon^{inel}}}{T}dt,$$
where t, $\dot{\varepsilon^{inel}}$, and T represent the time, inelastic deformation increment, and temperature, respectively. The inelastic deformation increment is computed by summing the non-recoverable strain increment and the dashpot deformation increment of the Maxwell elements.

The damage variable, D, can be calculated using Equation (6) based on the entropy generation per unit volume, s, and the final FFE of the material, $s^{cr}$.
(6)$$D=s\frac{D^{cr}}{s^{cr}}a^{d},$$

In this study, a time acceleration factor, $a^{d}$ is introduced based on previous research [27] to simulate fractures with fewer loading cycles and reduce the computational expense. In addition, a critical damage threshold, $D_{cr}$ is included, enabling the detection of failures before the entropy generation reaches the original FFE and, thus, accelerating the analytical process.

### 2.2. Time–Temperature Superposition Principle

Polymeric materials commonly adhere to the time–temperature superposition principle (TTSP) [39]. Utilizing the Arrhenius equation, Equation (7) is employed to model the behavior of the resin.
(7)$$\log\alpha^{TR}\left(T^{test}\right)=\frac{\Delta H}{2.303R}\left(\frac{1}{T^{test}}-\frac{1}{T^{ref}}\right),$$
where $\alpha^{TR}$, ΔH, R, $T^{ref}$, and $T^{test}$ represent the shift factor, activation energy, gas constant 8.314 $\times 10^{-3}$ kJ/(K mol), reference temperature, and testing temperature, respectively. The material properties of the Maxwell elements are determined at the reference temperature. Since the simulation accounts for temperature elevation, it is necessary to simulate the resin behavior at the testing temperature. The shift factor is used to relate the properties between the reference and test temperatures. In this study, the activation energy and reference temperature are set to 150 kJ/mol and 300 K, respectively. The relationship between the times at the test and reference temperatures is expressed in Equation (8).
(8)$$\tau \left(T^{test}\right)=\tau \left(T^{ref}\right)\times \alpha^{TR}\left(T^{test}\right),$$

The time-dependent flow is represented by the calculation of the stress in the dashpot of the Maxwell model. The stress in the dashpot is computed using the viscosity coefficient, η, and the strain increment in the dashpot, dε, as shown in Equation (9).
(9)$$\sigma =\eta \frac{d\varepsilon }{dt},$$

Based on Equations (8) and (9), the stress at the test temperature is calculated as in Equation (10).
(10)$$\sigma =\eta \frac{d\varepsilon }{dt\left(T^{ref}\right)}=\eta \frac{d\varepsilon }{\frac{dt\left(T^{test}\right)}{\alpha^{TR}}}=\alpha^{TR}\left(T^{test}\right)\eta \frac{d\varepsilon }{dt\left(T^{test}\right)},$$

Equation (10) shows that the temperature elevation is accounted for by multiplying the viscosity coefficient η by the shift factor $\alpha^{TR}$. As the temperature rises, the shift factor decreases, accelerating the movement of the dashpot and leading to increased energy dissipation. In other words, the temperature elevation causes an exponential increase in the dissipated energy, subsequently enhancing entropy generation. This increase in entropy generation accelerates the reduction in the material strength.

### 2.3. Heat Release

Heat release is calculated based on the heat flux across a surface. The relationship between the reference sink temperature ($\theta^{0}$), surface temperature (θ), and heat flux across the surface (q) is given by Equation (11).
(11)$$q=-h\left(\theta -\theta^{0}\right),$$
where h represents the surface heat transfer coefficient, set to 10 W/m^2^K in this study [40]. The heat flux q is directly related to the dissipated energy, which is derived from the inelastic deformation calculated using Equation (5). This dissipated energy is treated as heat energy and contributes to the temperature rise.

## 3. Finite Element (FE) Simulation

### 3.1. UMAT Calculation Flow

Figure 4 presents the calculation flowchart for updating the stress while considering the damage, where the subscript n refers to the nth Maxwell element. The stress is calculated using the user subroutine UMAT in Abaqus. Initially, the total strain increment (Δε), damage variable (D), damage increment (ΔD), previous stress ($\sigma^{old}$), and previous stress for the nth Maxwell element ($\sigma_{n}^{old}$) are input into UMAT. Next, the non-recoverable strain increment ($\Delta \varepsilon^{nr}$) and nonlinearity (g) are computed based on Equations (1) and (4), respectively. The strain increment for the Maxwell element $\Delta \varepsilon^{ve}$ is calculated as the difference between the total and non-recoverable strain increments. The temporary stress increment (${\Delta \sigma }^{temp}$) is obtained by taking the inner product of the total Young’s modulus ($E_{0}$) and $\Delta \varepsilon^{ve}$. The temporary stress for each Maxwell element ($\sigma_{n}^{temp}$) is calculated using $\sigma_{n}^{old}$ and distributing $\Delta \sigma^{temp}$ equally among all Maxwell elements. The dashpot elongation increment ($\Delta b_{n}$) is computed based on the transformed version of Equation (9), with g applied as a multiplier to increase the dashpot elongation. The temporary spring elongation increment ($\Delta a^{temp}$) is determined by dividing $\Delta \sigma^{temp}$ by $E_{0}$. The spring elongation increment ($\Delta a_{n}$) is then determined as the difference between $\Delta a^{temp}$ and $\Delta b_{n}$. The stress increment ($\Delta \sigma_{n}$) is subsequently computed, incorporating D. Following this, the stress is updated, and the entropy generation is calculated based on the stress and dashpot elongation increment, taking into account both the dashpot elongation and non-recoverable strain. Finally, the damage is calculated using Equation (6), and D is updated.

### 3.2. Material Properties

Table 1 lists the material properties, including those of the Maxwell elements and damage variables. The material properties of the Maxwell elements were utilized based on our previous study [41]. The fitting parameters for the non-recoverable strain were determined via a tensile test conducted on epoxy resin. A mixture of 6.3 g of 4,4′-diaminodiphenyl sulfone (4,4′-DDS) and 50 g of diglycidyl ether of bisphenol A (DGEBA) was stirred at 100 °C for 10 min and degassed four times at 80 °C. The mixture was then cured at 180 °C for 2 h, and test specimens with dimensions of 100 mm × 15 mm × 3 mm were prepared. The specimens were subjected to tensile loadings at a speed of 0.06 mm/s. Figure 5 presents the stress–strain curves determined from experimental testing and the simulation results (based on the flowchart in Figure 4). The parameters for the non-recoverable strain and nonlinearity used in the simulation were calibrated accordingly. The critical damage threshold, $D_{cr}$ used in this study was set at 0.25, with the resin considered to have failed upon reaching this value. This threshold was selected based on previous studies [14,26,28] that have used this criterion to simulate resin failure.

### 3.3. Simulation Model

The cyclic loading behavior is simulated using Abaqus 2020/Standard. As shown in Figure 6, A single coupled temperature-displacement element representing the resin, with dimensions of 1 mm × 1 mm × 1 mm, was simulated. Boundary conditions are applied to surfaces ADHE, ABCD, and DCGH, with X-, Y-, and Z-symmetry imposed, respectively; and heat transfer conditions are applied to surfaces ABFE, BCGF, and EFGH. The stress is updated using the UMAT subroutine, and the heat generation is calculated using HETVAL in Abaqus. Displacement boundary conditions are applied to surface EFGH at frequencies of 0.1, 0.25, 0.5, 1, 2, 4, 5, 6.25, and 10 Hz. The displacement is set to 0.01 mm, and the loading is cycled to induce fracturing.

## 4. Results and Discussion

Figure 7 shows the relationship between the loading frequency and the number of loading cycles until failure, obtained from the simulation in this study. At 0.1 Hz, the number of cycles to failure is notably low due to the significant progression of fatigue damage within each cycle. This is caused by the viscoelastic nature of the resin in CFRPs, where prolonged stress exposure at low frequencies accelerates damage accumulation, resulting in fewer cycles before failure [14]. At low frequencies, the dissipated energy due to viscoelasticity is greater than that due to non-recoverable strain per cycle. Consequently, viscoelastic behavior primarily contributes to the calculated dissipated energy, leading to increased damage progression at lower frequencies. As the frequency increases to 2 Hz, the number of loading cycles required to reach failure also rises, suggesting that in this frequency range, the material experiences less damage per cycle. However, when the frequency exceeds 2 Hz, the trend is reversed, and the number of cycles until failure decreases as the frequency increases; this accelerates damage progression as the temperature rises [15], ultimately leading to a reduction in the number of cycles to failure [11,16,17,18,19]. This result represents a significant achievement compared to previous analyses available in the literature. Earlier studies, which applied an entropy criterion incorporating only the matrix viscoelasticity, were unable to replicate this relationship, where the number of cycles to failure decreases at high frequencies and continues to decrease as the frequency increases. Including a time-independent non-recoverable strain term, in addition to the viscoelastic properties within the fatigue simulation framework of the current model, allowed for capturing more realistic frequency dependency effects in this study. This term accounts for the permanent deformation that accumulates regardless of time, enhancing the ability of the simulation to reflect actual material behavior.

Figure 8 presents the temperature changes during the simulations for loading frequencies of 0.1, 2, 6.25, and 10 Hz. The temperature increase is smallest at 0.1 Hz, and the higher the loading frequency, the greater the temperature rise observed. At 0.1 Hz, the temperature shows a slight initial increase but remains relatively stable throughout the cycles, indicating minimal heat generation from the slower loading rate. At 2 and 6.25 Hz, the temperature increases more substantially compared to the case at 0.1 Hz. However, after peaking, the temperature starts to decrease as the number of cycles increases. At 10 Hz, the behavior is markedly different, with the temperature continuing to rise throughout the loading cycles. Ultimately, the specimen breaks while the temperature is still rising, indicating that the high temperature contributes to accelerated material degradation and failure.

Figure 9 illustrates the relationship between the damage variable and the number of loading cycles for loading frequencies of 0.1, 2, 6.25, and 10 Hz. The resin is considered to have failed when the damage variable reaches 0.25. At 0.1 Hz, the slower loading rate allows for greater inelastic strain and larger increments of dissipated energy per cycle. As a result, the damage variable increases significantly from the start, indicating faster damage accumulation and leading to earlier failure at this low frequency. On the other hand, for frequencies higher than 2 Hz, the damage variable increases more rapidly as the frequency rises. At higher frequencies, the heat generated by cyclic loading accelerates the movement of the dashpot (per Equation (10)), resulting in larger increments of dissipated energy. This leads to faster damage accumulation, particularly when heat generation outweighs the effects of temperature changes, further increasing entropy generation and contributing to earlier failure. These results demonstrate that the relationship between the loading frequencies and the number of loading cycles to failure is closely tied to both the non-recoverable strain and temperature.

The trend observed in the simulation results, which is the frequency-dependent variation in the number of cycles until failure, aligns qualitatively with experimental observations (Figure 1), supporting the validity of the proposed model. Although this study has demonstrated the accurate simulation of fatigue behavior under various loading frequencies, some challenges remain that warrant future investigation. One of the key areas for future work is improving the accuracy of the non-recoverable strain representation. Since this term involves fitting constants, properly determining these constants is essential for enhancing the precision of strain measurement and modeling and thereby improving the reliability of fatigue predictions. Based on the findings of this study, an analysis for composite laminates such as CFRPs will be conducted, incorporating Hashin’s failure criterion. Implementing this criterion would provide a more comprehensive framework for predicting failure in complex composite structures.

## 5. Conclusions

The cyclic loading behavior of epoxy resin at different frequencies was simulated, where the number of cycles until failure in CFRPs varied depending on the loading frequency. To accurately capture this behavior, it is crucial to simulate epoxy resin, a key matrix resin in CFRPs. The main factors contributing to resin damage are inelastic deformation and temperature rises. Entropy generation, calculated from the dissipated energy associated with both the inelastic deformation and temperature, serves as the criterion for assessing damage. At low frequencies, the inelastic deformation per cycle is significant, leading to more rapid damage progression. At higher frequencies, heat generation surpasses heat dissipation, causing the inelastic deformation to increase according to the time–temperature superposition principle, which accelerates damage evolution. The key advantage of this study is the incorporation of both heat generation and non-recoverable strain, which are critical for accurately simulating the temperature rise at higher frequencies. By applying cyclic loadings to the epoxy resin at various frequencies, the proposed model effectively simulates the resin’s behavior until failure. This demonstrates that the proposed method not only accommodates different stress levels but also accounts for varying loading frequencies, making it highly adaptable for predicting resin behavior under diverse cyclic conditions.

## Figures and Tables

**Figure 1 materials-17-06202-f001:**
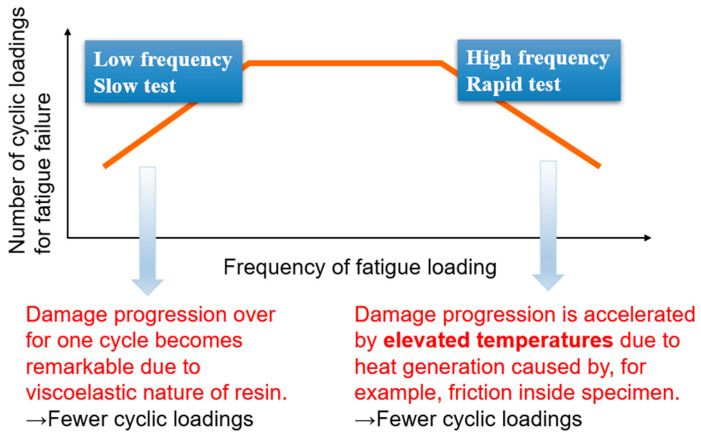
Relationship between fatigue and frequency: at low frequencies, fatigue failure is primarily caused by significant deformation, whereas at high frequencies, it is driven by temperature elevations.

**Figure 2 materials-17-06202-f002:**
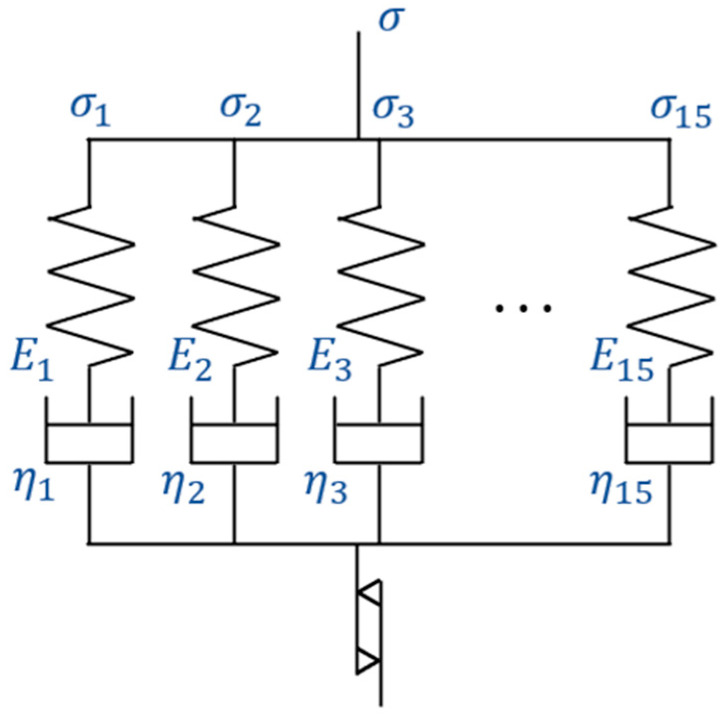
A viscoelastic–plastic model consisting of 15 parallel Maxwell elements and an additional plastic element.

**Figure 3 materials-17-06202-f003:**
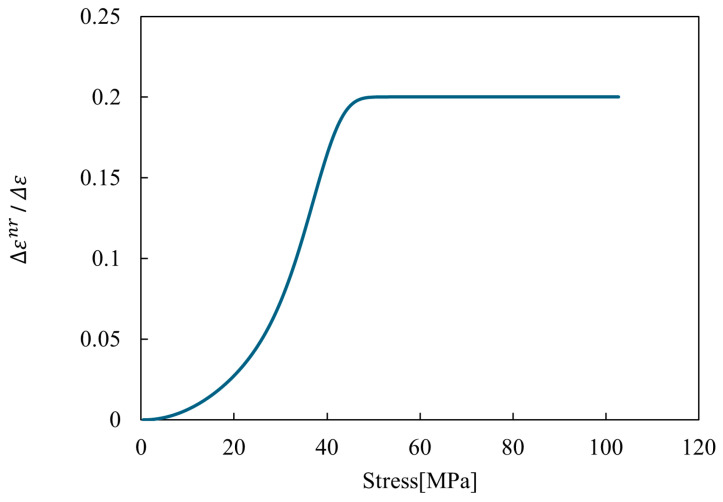
The relationship between stress and the non-recoverable strain increment, normalized by the total strain increment (Equation (1)).

**Figure 4 materials-17-06202-f004:**
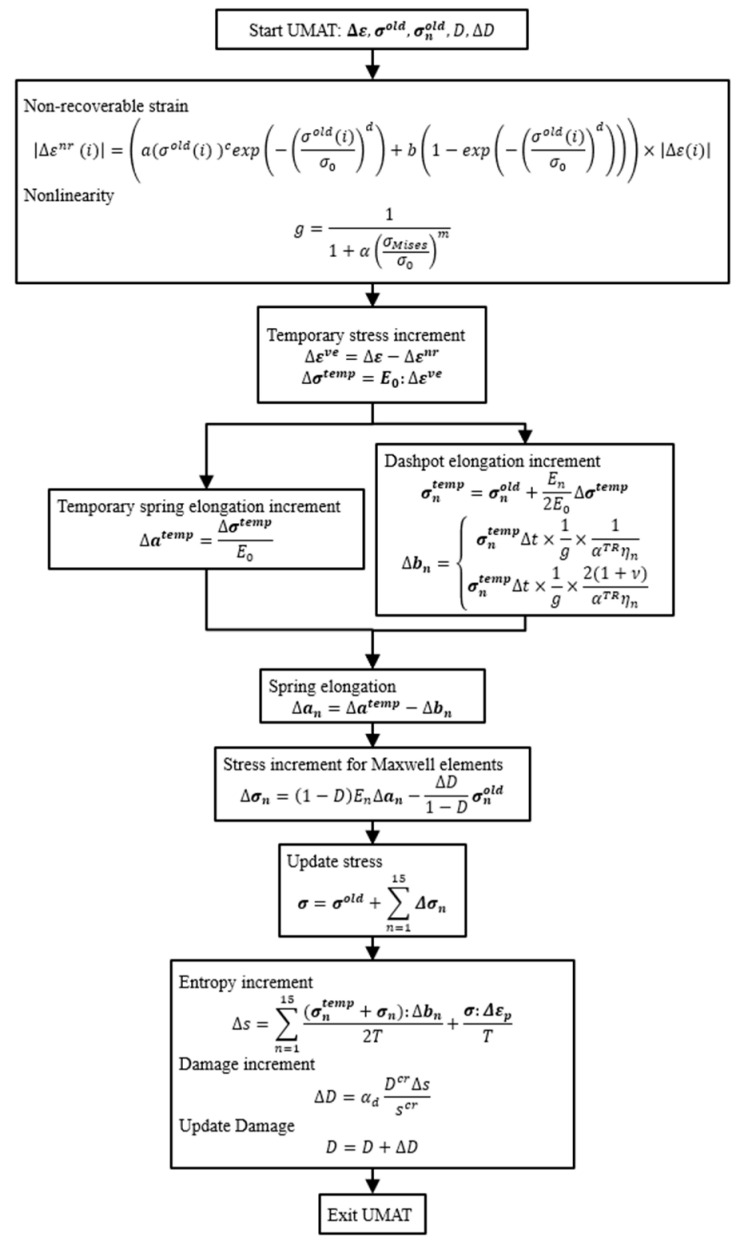
Calculation flowchart for the stress update in UMAT.

**Figure 5 materials-17-06202-f005:**
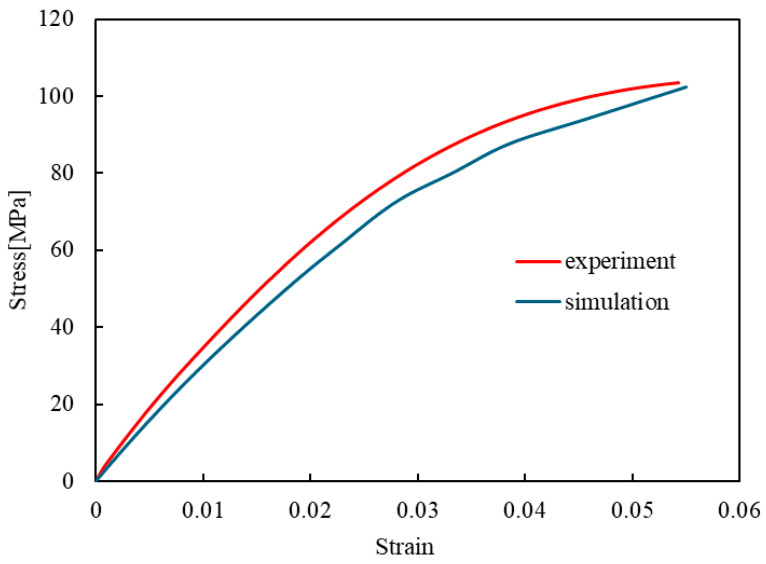
Stress–strain curves from the tensile test of the epoxy resin: experimental (red line) and FE simulation (blue line) results.

**Figure 6 materials-17-06202-f006:**
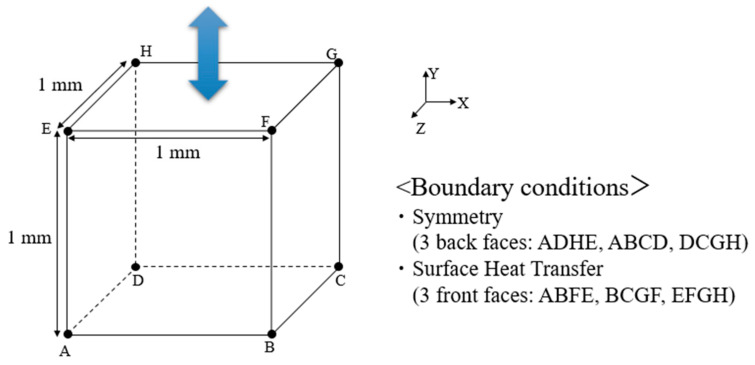
Boundary conditions applied to the element in the FE simulation.

**Figure 7 materials-17-06202-f007:**
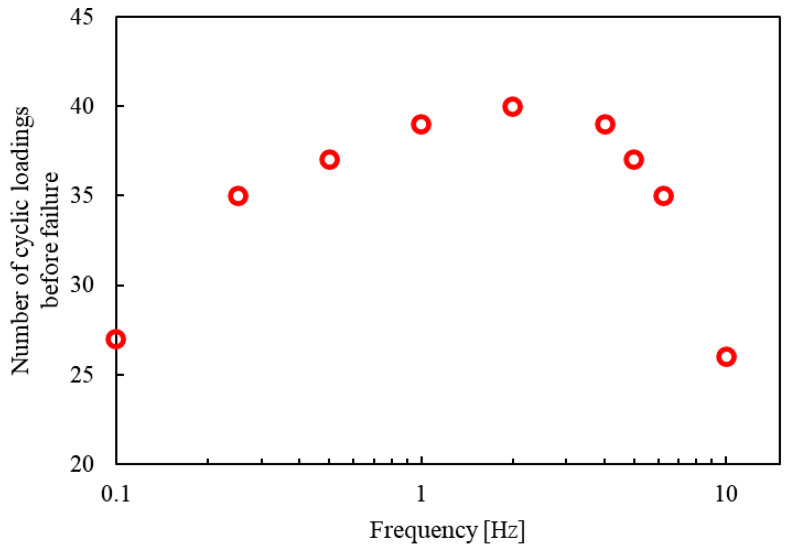
Relationship between the loading frequency and the number of loading cycles to failure.

**Figure 8 materials-17-06202-f008:**
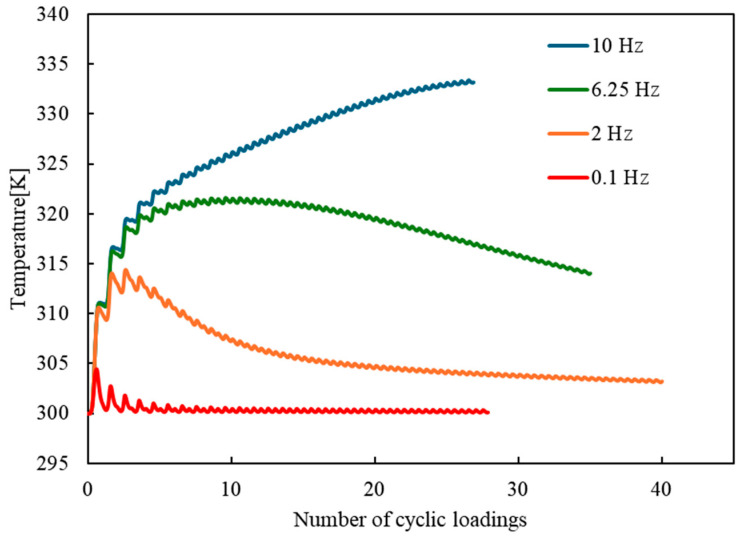
Temperature changes for loading frequencies of 0.1, 2, 6.25, and 10 Hz.

**Figure 9 materials-17-06202-f009:**
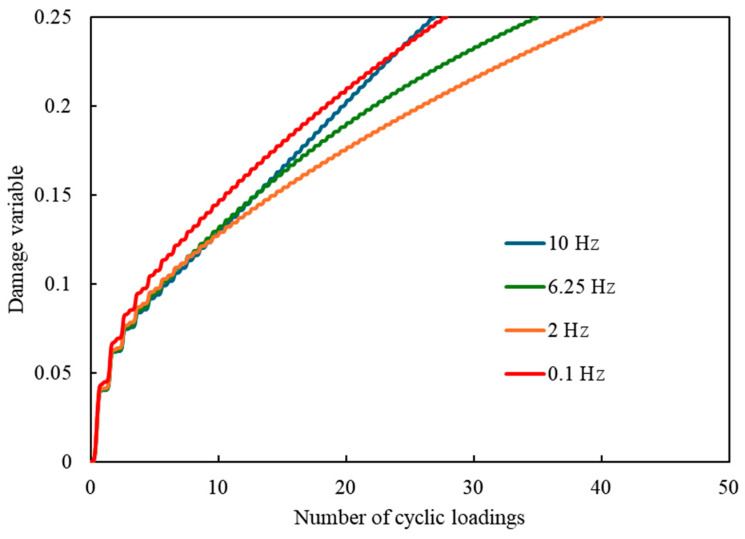
Relationship between the damage variable and the number of loading cycles for loading frequencies of 0.1, 2, 6.25, and 10 Hz.

**Table 1 materials-17-06202-t001:** Material properties, including those of the Maxwell elements and damage variables used in the FE simulation.

n	$E_{n}$ (MPa)	$\eta_{n}$ (MPa⋅s)	Elasticity	
1	220	2.51 × 10^3^	$E_{0}$ (MPa)	3300
2	220	6.31 × 10^5^	ν	0.34
3	220	1.58 × 10^7^	Nonlinearity	
4	220	6.31 × 10^7^	$\sigma_{0}$ (MPa)	60
5	220	7.94 × 10^8^	α	5
6	220	2.00 × 10^9^	m	18
7	220	7.94 × 10^9^	Non-recoverable strain	
8	220	3.16 × 10^10^	$\sigma_{0}$ (MPa)	40
9	220	1.58 × 10^11^	a	7.00 × 10^−5^
10	220	3.98 × 10^11^	b	0.2
11	220	2.00 × 10^12^	c	2
12	220	1.00 × 10^13^	d	8
13	220	7.94 × 10^13^	Damage variables	
14	220	2.00 × 10^15^	$\alpha_{d}$	10
15	220	1.00 × 10^35^	$D^{cr}$	0.25
			$s^{cr}$	30

## Data Availability

The original contributions presented in the study are included in the article, further inquiries can be directed to the corresponding author.

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
