# Peer review of "Numerical Simulation of the Frequency Dependence of Fatigue Failure for a Viscoelastic Medium Considering Internal Heat Generation"

_materials, 2024, doi:10.3390/ma17246202_

Round 1
Reviewer 1 Report
Comments and Suggestions for Authors
In this manuscript, authors proposed a novel damage evolution model to account for both non-recoverable strain as well as temperature effects. They used this model to analyze the effect of fatigue loading frequency on the fatigue life. The model description is comprehensive but some details still need improvement or clarify.
1) In abstract, Ln17-19 is not clear. "At low frequencies, ... , resulting in stable temperatures. Higher frequencies, ..., accelerated damage." Do you want to focus on temperature or damage? What do you mean by energy dissipation exceeds/balance heat release? How can you balance them if both are loss energy? Similar sentences across the manuscript need to be revised.
2) Figure 1, if you consider other fatigue loading factors, such as the stress/strain amplitude, mean stress, etc., what will change in this figure?
3) What is the rationale of building the model shown in Figure 2? For example, why do yo choose 15? What do you mean by "the 15 dashpots work across multiple timescales"? In which aspect did you considered multiple timescales?
4) Can you show a plot of eq (1)? i.e., non-recoverable strain v.s. stress? And perform some analysis on it?
5) Regarding heat release, eq (11) was introduced. However, how does it correlate the deformation or energy loss to temperature? Could you please elaborate it?
6) Figure 4 and Table 1, how did you calibrate so many parameters using a single stress-strain curve? Does the table -1 include all 15 elements? Can you explain more about the calibration procedure and the table itself?
7) What is your mesh type? How many finite elements in your model?
8) If it is a simple cube, with all symmetry boundary conditions, what is the point of using a complex finite element simulation? Is it possible to perform a simple 0-D analysis for the same purpose?
9) Figure 6, where does the figure come from? From literature or your own test, or your simulation?
10) Page 11, LN# 280. You mentioned the resin failed when damage reaches 0.25. This should be mentioned in earlier section. Also the question is, where does this magic number come from? Did you verify it?
11) The most important thing is that, did you perform the model verification and validation? What is the goodness of your proposed model if compare with the real experimental data?
12) You mentioned the advantage of the study, i.e., inclusion of inelastic strain and temperature. But what is the limitation of the model and your research overall?
Reviewer 2 Report
Comments and Suggestions for Authors
· in Figure 6, What role does the viscoelasticity of the resin in CFRPs play in fatigue damage at low frequencies?
· The last question is concerned with the function of temperature rise in enhancing damage evolution at high loading frequencies.
· In this study, why is the addition of the time-independent non-recoverable strain term beneficial to the fatigue simulation framework?
· How does this study’s model differ from earlier studies that employed an entropy criterion based only on matrix viscoelasticity?
· What implications does the result of this study have for designing or evaluating CFRPs under cyclic loading scenarios?
· Why might it be useful to know how the loading frequency affects fatigue failure in composite materials?
· What is the trend of the damage variable and the number of loading cycles in the case of different loading frequencies as represented in Figure 8?
· What is the relationship between the loading frequency and the increments of dissipated energy per cycle?
· At what loading frequency is the damage variable highest and why does the situation result in earlier failure?
· To what extent does the combined effect of non-recoverable strain and temperature affect the trend of loading frequencies and failure cycles?
· Which experimental settings or conditions would be most appropriate for the trends in Figure 8?
· As a reader, why did the authors refer to more than 10 citations for this paragraph? "The complexity of fatigue mechanisms in composite materials, including CFRPs, demands a reliable criterion for accurately estimating their lifetime. This has led to the development of various fatigue damage accumulation theories. Among these, the entropy-based approach has gained significant attention due to its ability to quantify irreversible degradation [14, 20-33]."
· Could you give a thorough review of the English language?
Reviewer 3 Report
Comments and Suggestions for Authors
An interesting paper.
The Authors could add the limitations of this theoretical study and future application of the obtained results.
Round 2
Reviewer 1 Report
Comments and Suggestions for Authors
The revised version of manuscript addressed my concerns.
Reviewer 2 Report
Comments and Suggestions for Authors
-